# Single Train Multi Deploy on Topology Search Spaces using Kshot-Hypernet

Jingyue Zhuge [1 2]   Christian Mayr [1 2 3]   Anand Subramoney [4]   David Kappel [5]

## Abstract

Neural Architecture Search (NAS) has long been an important research direction, to replace labor-intensive manual architecture search. Since the introduction of weight sharing in NAS, the resource and time consumption of architecture searches has been significantly reduced. In addition, variants of NAS methods have been proposed that eliminate the need for retraining by inferring model parameters directly from the shared weights after the search. However, these methods are mainly based on the MobileNet search space, which is primarily used for "size" searches. For the important "topology" search space, no NAS method has been proposed that does not require retraining. In this work, we fill this gap by proposing a NAS method that does not require retraining based on the topology search space. Our method combines the advantages of previously proposed Hypernetwork and Kshot-NAS. We also propose a new distillation and sampling method for this new NAS architecture. We present results on NAS-Bench-201 and show that our method matches or even exceeds the baseline performance of post-search retraining.

## 1. Introduction

Since its inception, Neural Architecture Search (NAS) has received much attention due to the challenges of manually designing neural network architectures. The original goal of NAS was to automatically design the optimal neural network architecture to solve specific tasks. The initial works (Zoph

[1]*Chair of Highly-Parallel VLSI-Systems and Neuro-Microelectronics, TU Dresden*, Dresden, Germany [2]Center for Scalable Data Analytics and Artificial Intelligence (ScaDS.AI), TU Dresden, Dresden, Germany [3]Centre for Tactile Internet (CeTI) with Human-in-the-Loop, TU Dresden, Dresden, Germany [4]Dept. of Computer Science, Royal Holloway, University of London, Egham, United Kingdom [5]Ruhr University Bochum, Institut für Neuroinformatik, Germany. Correspondence to: Jingyue Zhuge <jingyue.zhuge@tu-dresden.de>.

Accepted to the Workshop on Advancing Neural Network Training at International Conference on Machine Learning (WANT@ICML 2024).

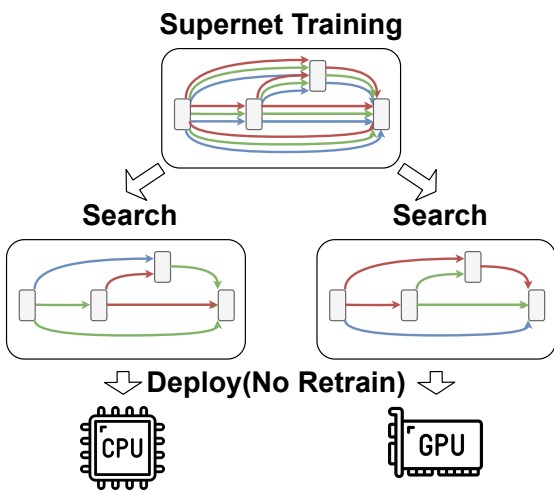

**Supernet Training**

**Search**          **Search**

**Deploy(No Retrain)**

CPU          GPU

*Figure 1.* Only the topology search space based supernetwork is trained. All subnets can be directly sampled from the supernetwork, and directly deployed, without the need to train from scratch.

& Le, 2017; Real et al., 2019; Zoph et al., 2018) were all based on this, and demonstrated their effectiveness at great cost, as they required training many candidate architectures from scratch for evaluation. After ENAS (Pham et al., 2018) proposed weight sharing, the efficiency and resource consumption of NAS were greatly optimized. As the demand for deploying neural networks on different platforms or different user devices increases, the goal of NAS has gradually expanded to designing neural network architectures suitable for different platforms. Due to the different characteristics of different platforms, the running speed of the same network architecture on different platforms will vary greatly (Wu et al., 2019). Therefore, for efficient deployment, it is usually necessary to repeat the search for each platform. If the architectures for each platform need to be trained from scratch, the computational cost of the NAS will be very high. Therefore, some works (Cai et al., 2020; Yu et al., 2020a) have proposed to use weight sharing to train only one large supernetwork, and all the weights of the subnetworks can be sampled directly from the supernetwork. In this way, the architecture can be directly sampled from the weights of the

supernetwork for verification and deployment, without the need to train from scratch.

However, the current methods are based on *size* search spaces, such as the MobileNet search space (Howard et al., 2019), which is different from the previous NAS (Zoph & Le, 2017; Real et al., 2019; Zoph et al., 2018) based on *topology* search spaces (Directed Acyclic Graph). Although the size search space is also considerably large that can be effectively searched for different devices. In topology search space, the architecture or search block, is defined by the number of nodes in the DAG and the edges between nodes, as well as the candidate operations. The network is stacked from search blocks. We can search for one architecture for each block, or all search blocks can use the same architecture. In contrast, the blocks in the size search space are predetermined, and the network is stacked from multiple blocks with the same architecture. For example, in the MobileNet search space, the block is the inverted bottleneck block, and the structure of this block is fixed. It consists of two point-wise convolutions at the beginning and end for channel scaling, and a depth-wise convolution in the middle for feature extraction. The variable parts are the number of channels in the block (including the expansion rate), the kernel size of the depth-wise convolution, and the depth of the entire network. The limitation of this search space makes the resultant architectures be the variants of MobileNet, rather than completely new architectures. In other words, smaller subnets can be fully contained by larger subnets. In other words, the main drawback is that all subnets have the same architecture, and the performance is mainly affected not by the differences in architecture, but by the number of FLOPs. For deployment, the constraints mostly come from the size of hardware storage and computational capabilities. In contrast, the operations in the search block of topology search space do not intersect. This leads to very large differences in the architectures of different subnets. For hardware with specific accelerators, this can make NAS more meaningful. However, this feature makes supernetwork training more difficult. Therefore, current work based on topology search spaces (Su et al., 2021; Zhao et al., 2021; Hu et al., 2020) aims to improve the ability of ranking based on supernetwork weight sharing. But, the searched network still needs to be retrained from scratch to achieve the deployable performance.

Here, we investigate the problem of NAS based on topology search spaces always needing to be retrained to achieve deployable performance. We propose a supernetwork training method that combines multiple techniques to solve this problem. In terms of network structure, we integrate Hypernetwork and KshotNAS, enabling the network weights to adapt based on the architecture. In terms of training, we introduce a novel distillation approach combined with Focus-Fair Sampling. This allows all subnets to be rela-

tively well-trained without the need to train from scratch, as shown in Figure 1. We conducted experiments on NAS-Bench-201 (Dong & Yang, 2020). On Cifar10, we achieved an average accuracy of 87.12% averaged across all subnets (the equivalent value for training from scratch is 87.06%), while the best accuracy was 92.47% (the value for training from scratch is 94.37%). On Cifar100, the average accuracy of all subnets reached 61.03% (the value for training from scratch is 61.41%), and the best accuracy was 71.98% (the equivalent value of training from scratch is 73.51%).

## 2. Related Work

Conventional NAS (Zoph & Le, 2017; Real et al., 2019; Zoph et al., 2018; Liu et al., 2018) requires training a large number of candidate architectures from scratch to select the best-performing architecture, which consumes a lot of computational resources and time, hindering the widespread application of NAS. The proposal of weight sharing, that is, using a supernetwork that includes all architectures in the search space, only trains the supernetwork, and all subnetworks can be obtained by sampling the weights of the supernetwork, greatly reduces the computational and time costs compared to traditional NAS. ENAS (Pham et al., 2018) reduces the cost by 1000x by using weight sharing. Differentiable Architecture Search(DARTS) (Liu et al., 2019) adds coefficients to each path, and the output of all nodes is the weighted sum of all paths. By this technique, the search method is made differentiable, and the optimal architecture is searched using gradient descent. ProxylessNAS (Cai et al., 2019) adds additional architectural parameters during training and uses binary encoding to activate only one path at a time.

**Sampling Method:** During the training of the supernetwork, each iteration needs to sample the subnetwork, and the sampling method will have a non-negligible impact on the final training results of the supernetwork. OneShot-NAS (Bender et al., 2018) discards paths during training, and the discard rate increases over time. Single Path One-Shot(SPOS) (Guo et al., 2020) compresses the search space so that all subnetworks are single-path, and a random path is selected at each iteration, so the supernetwork is just a framework and does not need to be fully trained. FairNAS (Chu et al., 2021) uses a fair sampling path method and superimposes gradients for simultaneous updates to eliminate the SPOS subnetwork iteration order problem, reducing the optimization gap between subnetworks. DFairNAS (Meng & Chen, 2023) improves on FairNAS, proposing a way to calculate the score of all operations based on the performance of the subnetworks to combine high-scoring operations. This is consistent with our idea, we propose a new FocusFair sampling method, which has a higher probability of sampling high-performance subnetworks while minimizing the

impact on the remaining subnetworks.

**Size Search Space based NAS:** Single-Path NAS (Stamoulis et al., 2019) uses the MobileNet search space and proposes sharing of convolutional kernels, where small convolutional kernels inherit from large convolutional kernels. Once-for-All(OFA) (Cai et al., 2020) performs progressive shrinkage fine-tuning on the supernetwork after full training, so that the subnetworks can also be directly deployed without retraining. BigNas (Yu et al., 2020a) uses many techniques during training, such as the sandwich rule (Yu & Huang, 2019), inplace distillation (Yu & Huang, 2019), and exponentially decaying with constant ending, to achieve almost the same effect as OFA (Cai et al., 2020). However, as mentioned earlier, these are all based on the small search space and will not search for new architectures, all of which are variants of MobileNet, but the methods proposed are very meaningful. For example, in-place distillation, and OFA (Cai et al., 2020) proposed a transformation matrix for convolutional kernel transformation. We extend in-place distillation to the topological search space, effectively improving the training effect of the supernetwork. The purpose of the transformation matrix is to ensure that the weights between subnetworks are not completely shared, enhance the representation ability of the supernetwork, and have the same idea as our use of Hypernetwork. Autoformer (Chen et al., 2021) proposes the concept of weight entanglement and extends it from CNN to transformers. There are also some similar works that do not require retraining, such as Hardware-Aware Transformers (HAT) (Wang et al., 2020), Focusformer (Liu et al., 2022a), AttentiveNAS (Wang et al., 2021), NASVIT (Gong et al., 2022), ShiftNAS (Zhang et al., 2023).

**Rank Correlation:** Weight sharing has not been theoretically verified, only based on an assumption that the subnetwork ranking obtained by evaluating the subnetwork trained by the supernetwork is valid. A large number of works are dedicated to verifying or improving the correlation between the ranking obtained by evaluating the subnetwork trained using the supernetwork and the ranking obtained by evaluating the subnetwork trained from scratch. (Hu et al., 2020) proposed a angle based method to shrink the search space to improve the ranking correlation. The work of (Zhang et al., 2020b) shows that the ranking correlation based on weight sharing is unstable, the reason being the mutual interference between a large number of subnetworks. Their study on group sharing shows that grouping based on network architecture similarity can effectively reduce the number of subnetworks while obtaining better ranking correlation. FewShot (Zhao et al., 2021) conducted the same study, extending OneShot-NAS to FewShot-NAS, which means that it has several supernetworks instead of one. Consistent with the results of (Zhang et al., 2020b), the more supernetworks there are, the better the ranking correlation. Another work

(Liu et al., 2022b) based on FewShot-NAS, changed the grouping method so that the number of groups gradually increased. KShot-NAS (Su et al., 2021) directly uses $K$ weights for each Conv layer, and introduces simplexnet, which encodes the network architecture as input and outputs $K$ weight coefficients, and finally obtains the network by summing the weighted $K$ weights. Through our method, all networks can be trained to a directly deployable state, which makes ranking correlation no longer critical.

**Hypernetwork based NAS:** There are two works (Brock et al., 2017; Zhang et al., 2020a) that use Hypernetwork (Ha et al., 2016) similarly to our method to generate the weights of the network. Smash (Brock et al., 2017) encodes the network architecture as a 3D tensor as input, and uses a 26-layer DenseNet (Huang et al., 2018) to generate all the weights of the network at once. Graph Hypernetworks(GHN) (Zhang et al., 2020a) encodes the network architecture as a computational graph to generate weights. The advantage of Hypernetwork in the NAS field is that it can generate weights based on the input architecture encoding, no longer completely weight sharing, which can improve the performance that subnetworks can achieve to a certain extent.

## 3. Method

Unlike the previous methods without retraining, we make it possible to perform multiple searches and deployments after training the supernetwork based on the topology search space once. Our method consists of two main parts:

- We combine Hypernetwork (Ha et al., 2016) and Kshot-NAS (Su et al., 2021) to make the expression ability of Hypernetwork more powerful, so that the weights of the architectures in the search space are no longer completely shared, and higher performance can be achieved.

- Based on the topology search space, we propose a new supernetwork training process, which includes distillation and Focus-Fair sampling methods.

We will first introduce the method of combining Hypernetwork and KshotNAS in Section 3.1, and then introduce the distillation and sampling methods in Section 3.2.

### 3.1. Kshot-Hypernet

Kshot-Hypernet is a combination of Hypernetwork(Ha et al., 2016) and KshotNAS(Su et al., 2021). The core of the Hypernetwork method is weight decomposition, which decomposes the weight of the convolution into the form of two matrices multiplied together, and all convolution layers share the same Hypernetwork. This method can greatly reduce the number of parameters. Specifically, we assume that

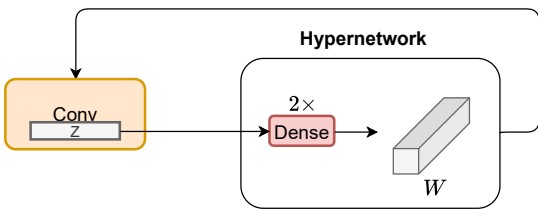

*Figure 2.* Hypernetwork as proposed in (Ha et al., 2016)

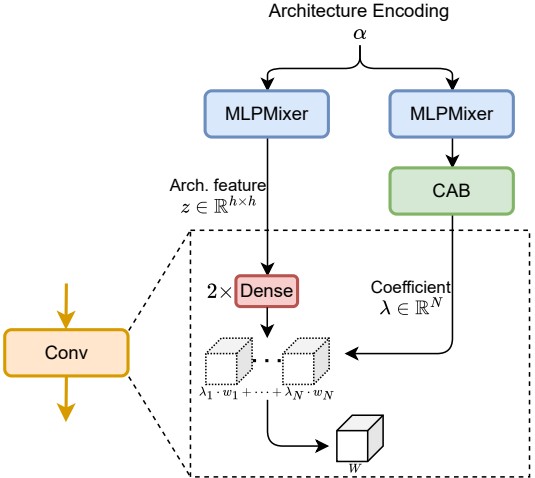

*Figure 3.* Kshot-Hypernet

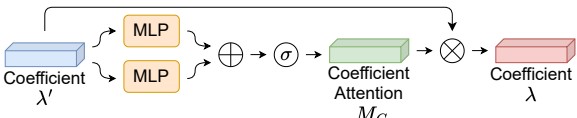

*Figure 4.* Coefficient Attention Block

a convolutional neural network has $D$ convolution layers. The weight of the $i$-th layer is $W^i \in \mathbb{R}^{C_{in}^i \times C_{out}^i \times K^i \times K^i}$, $C_{in}^i, C_{out}^i$ are the corresponding input and output channel numbers, and $K$ is the convolution kernel size. In order to allow the shared Hypernetwork to generate different weights, an additional learnable feature vector $z^i \in \mathbb{R}^h$ is added to each layer, where $h$ is the hidden dimension. The weight can be represented as follows:

$$W^i = g(z^i) ,\tag{1}$$

where $g$ is a generative function composed of two fully connected layers, as shown in Figure 2, with weight dimensions of $h \times C_{in}^i \cdot h$ and $h \times C_{out}^i \cdot K^i \cdot K^i$. Since the number of input and output channels of each convolution layer may be different, the output of the Hypernetwork is fixed to a unit convolution kernel, such as $16 \times 16 \times 3 \times 3$, and then multiple unit convolution kernels are generated and stacked.

Although the original Hypernetwork greatly reduces the number of parameters, its impact on network performance

cannot be ignored. To balance the number of parameters and performance, we allow each convolution layer to have its own weight generation network. The original method shared the weight generation network, but the trainable feature vectors of each unit convolution kernel were independent. In contrast, our weight generation network is independent, and the feature vectors are shared. The feature vectors are generated by the architecture encoding through a separate additional small feedforward network, which provides a superior efficiency accuracy tradeoff for NAS. The feedforward network consists of multiple layers of MLPMixer (Tolstikhin et al., 2021). To further reduce the number of parameters, we increase the shared part, i.e., the generated architecture features, and decrease the independent part, i.e., the weight generation network. The new weight can be represented as follows:

$$W^i = g^i(z) ,\tag{2}$$

where $z \in \mathbb{R}^{h \times h}$ is the feature vector generated by the architecture encoding, and the weight dimensions of the two fully connected layers of $g^i$ are $h \times C_{in}^i$ and $h \times C_{out}^i \cdot K^i \cdot K^i$, respectively.

However, even with this modification, the expressive power of Hypernetwork is still limited. To further enhance the expressiveness, we combine the method of KshotNAS with the weight representation Eq. (2).

The increase in the number of weights $N$ in the weight dictionary in KshotNAS, which leads to a surge in the number of parameters, makes network training more difficult (Su et al., 2021). One of the biggest advantages of Hypernetwork is the smaller number of parameters. The combination of the two allows us to use a larger $N$ without affecting the training of the network, while improving the expressiveness of Hypernetwork. In our method, the increment of the number of parameters is based on the number of weights in the first fully connected layer of the weight generation network, i.e., $h \times C_{in}^i \cdot N$. The weight dictionary can be represented as follows:

$$\Theta_{W^i} = [w_1^i, ..., w_N^i] = g^i(z) ,\tag{3}$$

where $w_n^i \in \mathbb{R}^{C_{out}^i \times C_{in}^i \times K^i \times K^i}$. Assuming that the input and output channels $C_{in}$ and $C_{out}$ of a convolution layer are both 256, the convolution kernel $K$ size is 3, the hidden dimension $h$ is 64, and the number of generated weights $N$ is 64. Then the original method requires

$C_{out} \cdot C_{out} \cdot K^2 \cdot N = 37.75M$, while our method only requires $C_{in} \cdot h \cdot N + C_{out} \cdot K^2 = 1.05M$. The number of parameters is reduced by almost $35\times$. Moreover, K-shot uses SimplexNet to generate weight coefficients, which is similar to the shared part of our network. Therefore, we combine the two to generate weight coefficients and architecture features at the same time. Thus, we can generate different weight dictionaries and their coefficients for different architectures, greatly increasing the expressiveness of the weight dictionary. The final weight can be represented as follows:

$$W^i = \sum_{n=1}^{N} \lambda_n w_n^i \ , \qquad (4)$$

$$\lambda = softmax(f(\alpha)) \ , \qquad (5)$$

where $\lambda \in \mathbb{R}^N$ is the weight coefficient, $\alpha$ is the architecture encoding, and $f$ is a multi-layer MLPMixer. To further increase the performance of the network, we insert a modified channel attention module (CAM) (Woo et al., 2018) before the $softmax$, which was proposed for convolutional neural networks to perform self-attention on the channel dimension of the input features. We call it the Coefficient Attention Block (CAB) (see Figure 4).

### 3.2. Training a Topology Search Space Based Supernetwork

Training a supernetwork based on a topology search space is more difficult than training a supernetwork based on a size search space. In a size search space, such as the MobileNet search space, where the smallest subnetwork is the shared part of all networks, in other words, all subnetworks can actually be considered as the result of pruning the supernetwork to a certain extent. From this perspective, we can improve the original NAS process based on weight sharing, omit the retraining step, and perform progressive shrinking consistent with pruning (Cai et al., 2020). Another, more aggressive strategy is to train all architectures at once without any fine-tuning or training (Yu et al., 2020a). For a topology search space, there is no subordinate relationship between architectures, only partial overlap. This makes it more difficult to train all subnetworks at once. We propose our solution to the sampling method and distillation method in supernetwork training. Figure 5 shows our training process.

**Distillation in topology search space:** Knowledge distillation is a technique to improve network performance. BigNAS(Yu et al., 2020a) verified that using the largest subnetwork for in-place distillation(Yu & Huang, 2019) is feasible in a size search space, and the effect is significant. In NAS based on a topology search space, the architectures between subnetworks are very different, making uniform distillation very difficult. Although not absolute, larger networks, or networks with more parameters, usually have

better performance. For a topology search space, since adjacent two layers are single-path, the architecture with the most parameters does not include the other subnetworks, so the subnetwork with the most parameters is not necessarily suitable for distilling the other subnetworks. Intuitively, the entire supernetwork is the most suitable teacher, because the supernetwork contains all subnetworks, and will have some of their characteristics. In previous NAS methods, the supernetwork was just a framework and did not need to be fully trained, but if the entire supernetwork is to be trained, directly accumulating the outputs of all paths will cause the network to be not trainable. Inspired by DARTS(Liu et al., 2019), we use additional architecture parameters $\beta$ to balance the outputs of different operations:

$$x^j = \sum_{o \in \mathcal{O}^{i,j}} \frac{exp(\beta_o^{i,j})}{\sum_{o' \in \mathcal{O}^{i,j}} exp(\beta_{o'}^{i,j})} o^{i,j}(x^i) \ , \qquad (6)$$

where $\mathcal{O}^{i,j}$ is the set of all operations between node $x^i$ and node $x^j$. Since it includes all operations, it is more suitable as a teacher for all subnetworks than a single-path network. Unlike DARTS, the purpose of which is to search for the best architecture on the validation set, our purpose is to train the best teacher for all subnetworks. This is reasonable, and experiments have shown its effectiveness. Although this increases the training time and GPU storage usage, the results are very worthwhile.

In addition, we do not disable the parameters of the Batch-Norm layer in the network as DARTS did, although it will affect the scaling of the architecture parameters $\beta$. This is because 1) we are not training the supernetwork alone, but training the supernetwork and subnetworks simultaneously, so the parameters of the Batchnorm layer are helpful for the training of the subnetwork, and the training of the subnetwork can correct the parameters of the Batchnorm layer. 2) Our goal is to train the best teacher for all subnetworks, not to search for the best architecture on the validation set, so the scaling of $\beta$ will not have a significant impact on our goal.

As shown in Figure 5, for each batch of training samples, we first train with the teacher to obtain soft labels, update the weights of the entire network and the architecture parameters $\beta$. Then we sample the subnetworks, train with the same training samples, and only calculate the loss using soft labels. We have tried mixing distillation loss and target loss, but using only soft labels can achieve better results.

**FocusFair Sampling:** The sampling method also has a great impact on the training of weight-sharing NAS. As described in FairNAS(Chu et al., 2021), using uniform sampling has a sequence problem. FairNAS can alleviate this problem to some extent. (Yu et al., 2020b) verified that compared with uniform sampling, FairNAS has better stability in the training of the supernetwork, especially in the early stage.

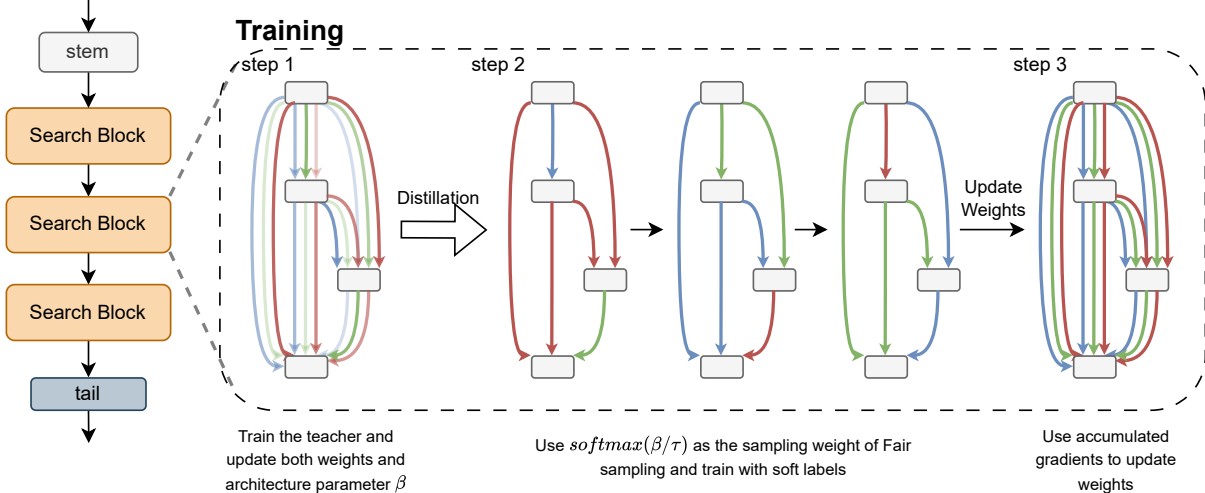

**Training**

*Figure 5.* Training flowchart. The first step is to train the complete supernetwork with architecture parameters, including forward and backward (the depth of color represents the size of the architecture parameters). The second step is to sample the subnetworks using the Focus-Fair sampling method based on the architecture parameters, and distill the subnetworks with the soft labels from the first step. Finally, the accumulated gradients are used to update the weights of the supernetwork.

In addition, as shown in Figure 6(a), through experiments, we found that for a topology search space, low-performance architectures are more likely to be positively affected in the training of the supernetwork, which get improvement to reach or exceed the performance of training from scratch. High-performance architectures will be affected more negatively, resulting in performance far worse than training from scratch. To address this problem, i.e., make the training more focused on high-performance architectures while minimizing the impact on other architectures, we propose Focus-Fair Sampling. Similar to the idea of DARTS(Liu et al., 2019), architectures with higher architecture parameters $\beta$ should have relatively high performance. Therefore, we use $softmax(\beta)$ as the sampling weight of FairNAS, so that operations with higher $\beta$ have a higher probability of being combined together, while still traversing all operations between each pair of nodes in each iteration. This ensures that the impact on architectures other than high-performance architectures is minimized. For example, assuming that the search block has three nodes, i.e., $[x^0, x^1, x^2]$, the operation pool is $[o_0, o_1, o_2]$, $softmax(\beta^{0,1}) = [0.2, 0.5, 0.3]$, and $softmax(\beta^{1,2}) = [0.3, 0.1, 0.6]$. Then the first sub-network sampling in the iteration has a $30\%$ probability of obtaining $[o_1^{0,1}, o_2^{1,2}]$, which is nearly twice higher than the uniform sampling $11.1\%$. Combined with our distillation method, this sampling method is cost-free. Compared with DFair(Meng & Chen, 2023), our method is simpler and does not require a large amount of validation during training.

To avoid the value of $max(softmax(\beta))$ being too large

during training, i.e., focusing too much on a single high-performance architecture, leading to training imbalance, we use an additional hyperparameter $\tau$, and we use $softmax(\beta/\tau)$ as the weight, which makes the sampling smoother. Through experiments, we found that the value of $\tau$ is best around $1.5$.

## 4. Result

In order to evaluate the overall performance of our method, that is, the performance of all architectures in the search space, we choose to use NAS-Bench-201(Dong & Yang, 2020) for testing. NAS-Bench-201 is a NAS benchmark based on cell search. The search space is the topology structure, that is, the directed acyclic graph (DAG). Each cell has four nodes, and there are five operations that can be selected between any two nodes, zeroing, skip connection, 1x1 convolution, 3x3 convolution, and 3x3 average pooling. The search space has a total of 15625 architectures. For all architectures, NAS-Bench-201 provides detailed data on training from scratch for 12 epochs and 200 epochs on three datasets, Cifar10, Cifar100, and ImageNet16-120. Therefore, NAS-Bench-201 is often used to evaluate the search ability of NAS methods and the ranking ability of weight sharing methods. Our goal is to evaluate the training effect of the supernetwork, that is, whether the subnet sampled from the supernetwork can directly achieve the performance of training from scratch. Therefore, we use the average accuracy of all subnets for evaluation.

---

**Algorithm 1** Training a Topology Search Space Based Supernetwork

---

**Input:** number of training epochs $E$, warmup epochs $E_w$, training data loader $D$, number of generated weights $N$, weight coefficients $\lambda$, operations pool $O$, number of operation candidates $K$.
**for** $e = 0$ **to** $E - 1$ **do**
  **if** $e < E_w$ **then**
    set $\lambda_n$ to $1/N$; #warmup phase
  **end if**
  train the entire supernetwork and get the soft label $\hat{y}$;
  calculate the loss with true label $y$ and backward pass;
  update weights and $\beta$ with weight decay;
  **for** $k = 0$ **to** $K - 1$ **do**
    random sample one architecture from $O$ with weights $softmax(\beta/\tau)$;
    remove sampled operation from $O$, and its corresponding architecture parameter from $\beta$;
    train the sampled architecture with soft label $\hat{y}$;
  **end for**
  update weights without weight decay;
**end for**

---

**Supernetwork Training:** For the configuration of the Hypernetwork, we use $h = 32, N = 64$. In order to make the architecture feature and weight coefficient generation network have a larger batch size at each update, we use the same trick as GHN3(Knyazev et al., 2023), that is, using the same input samples on each GPU, but training with different subnets. We use 4 GPUs for training. According to the setting of NAS-Bench-201, we sample 5 subnets each iteration (a total of 5 operations can be selected), so the batch size of the architecture feature and weight coefficient generation network each iteration is 20. Training uses SGD optimizer with 0.9 momentum and Nesterov acceleration. The batch size on each GPU is 256. We use an initial learning rate of 0.2, use cosine learning rate decay, and train for a total of 300 epochs. For Focus-Fair sampling, we set the temperature $\tau = 1.5$. In addition, we use the same warm-up method as KshotNAS(Su et al., 2021), that is, making all weight coefficients the same in the first 5 epochs. In-place distillation, we also use the same weight decay strategy as BigNAS(Yu et al., 2020a), that is, only weight decay is used for the teacher. We also remove weight decay for all BatchNorm layers and bias. Our complete training process is summarized in Algorithm 1. After the supernetwork training is completed, we do not perform any retraining and directly evaluate the subnet sampled from the supernetwork. During evaluation, we use the same method as (Yu et al., 2018), using 2048 training samples to recalculate the running statistics of the BatchNorm layer. To ensure fairness, our experiments are consistent with NAS-Bench-201, and no additional data augmentation is used.

Table 1 shows the training results of our method on NAS-Bench-201. Since other methods do not report the average accuracy, we cannot directly compare with them. Our method has a slightly higher average accuracy on Cifar10

*Table 1.* Training results on NAS-Bench-201.

|  | Cifar10 | Cifar100 |
|---|---|---|
| Avg. accuracy (baseline) | 87.06% | 61.41% |
| Avg. accuracy (ours) | 87.12% | 61.02% |
| Max. accuracy (baseline) | 94.37% | 73.51% |
| Max. accuracy (ours) | 92.47% | 72.04% |

*Table 2.* Effect of our Kshot-Hypernet based on Cifar100. The result of Kshot(Su et al., 2021) is produced by us, since they didn't provide source code.

|  | Avg. accuracy | Max. accuracy |
|---|---|---|
| weight sharing | 53.04% | 65.24% |
| Kshot(N=12) | 53.75% | 63.88% |
| Kshot-Hypernet | 60.66% | 68.4% |

than the baseline, and is very close to the baseline on Cifar100. The difference between the maximum accuracy and the baseline is kept within 2%. This indicates that our method can make most subnets on NAS-Bench-201 approach or surpass the results.

**Effectiveness of Kshot-Hypernet:** We compare the effect of directly using weight sharing, Kshot-NAS(Su et al., 2021), and our Kshot-Hypernet method on Cifar100. As shown in Table 2, our method has an improvement of about 7% in average accuracy and about 5% in maximum accuracy. This indicates that our method far exceeds other methods in the results of supernetwork training in the topology search space.

**Effectiveness of our distillation method:** To verify the effectiveness of our distillation method, we conducted an

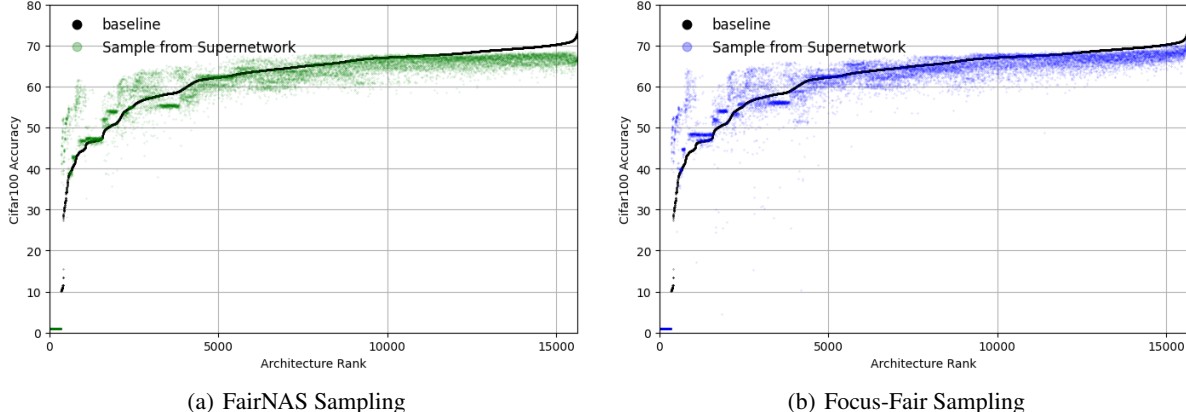

(a) FairNAS Sampling         (b) Focus-Fair Sampling

*Figure 6.* Comparison of FairNAS Sampling and Focus-Fair Sampling. The baseline is the result of training from scratch, provided by NAS-Bench-201.

*Table 3.* Effect of our distillation method based on Cifar100.

|  | Avg. accuracy | Max. accuracy |
|---|---|---|
| w/o KD | 58.60% | 67.86% |
| w/ our KD | 60.26% | 69.3% |

*Table 4.* Kendall's Tau of different methods on NAS-Bench-201.

| Method | Cifar10 | Cifar100 |
|---|---|---|
| SPOS (Guo et al., 2020) | 55.00% | 56.00% |
| AngleNet (Hu et al., 2020) | 57.48% | 60.40% |
| K-shot (Su et al., 2021) | 62.64% | 61.22% |
| FewShot(25-supernets) (Zhao et al., 2021) | 69.6% | N/A |
| our | 69.42% | 70.18% |

ablation experiment. As shown in Table 3, our distillation method has an improvement of about 1.5% in both average accuracy and maximum accuracy on Cifar100. This indicates that our distillation method is effective in improving the performance of the subnet, and overall improves the performance of the supernetwork.

**Effectiveness of Focus-Fair Sampling:** As mentioned in Section 3.2, previous sampling methods, such as FairNAS sampling, will result in high-performance subnets not getting enough training. To verify the effectiveness of our Focus-Fair sampling, we compared the training results of the two sampling methods. As shown in Figure 6, our method fills the gap in high-ranking subnets caused by FairNAS sampling.

**Ranking ability:** Although ranking ability is not our focus, we still evaluated the ranking ability of our method on NAS-Bench-201. As shown in Table 4, the Kendall's Tau

value of our method on both datasets is about 70%. This indicates that the ranking ability of our method is higher than almost all other methods. This also indirectly proves the effectiveness of our method.

## 5. Discussion

In this paper, we propose a novel training paradigm for super-networks based on the topological search space. After training the super-network, it can be directly searched and deployed on the target platform without the need for retraining. We introduce a new distillation and sampling method for topological search space NAS, which effectively improves the performance of all architectures in the search space after super-network training. Our method transcends the limitations of the MobileNet search space, enabling the training of a super-network to be applicable across various platforms, thereby increasing flexibility and efficiency in deployment.

Although we have achieved promising results on CIFAR-10 and CIFAR-100, there is still potential for improvement in maximum accuracy, necessitating further experiments. Additionally, it is essential to verify the method's effectiveness on larger datasets and broader search spaces. Future research may focus on integrating smaller search spaces into a unified training approach, thereby enhancing the flexibility of the NAS search space and improving deployment efficiency.

## Acknowledgements

Jingyue Zhuge is funded by Center for Scalable Data Analytics and Artificial Intelligence(ScaDS.AI) and the German Federal Ministry of Education and Research (BMBF) project EVENTS (16ME0733). David Kappel is funded by

the German Federal Ministry for Economic Affairs and Climate Action (BMWK) project ESCADE (01MN23004A). The authors gratefully acknowledge the computing time made available to them on the high-performance computer at the NHR Center of TU Dresden. This center is jointly supported by the Federal Ministry of Education and Research and the state governments participating in the NHR (www.nhr-verein.de/unsere-partner).

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

# A. Architecture distribution

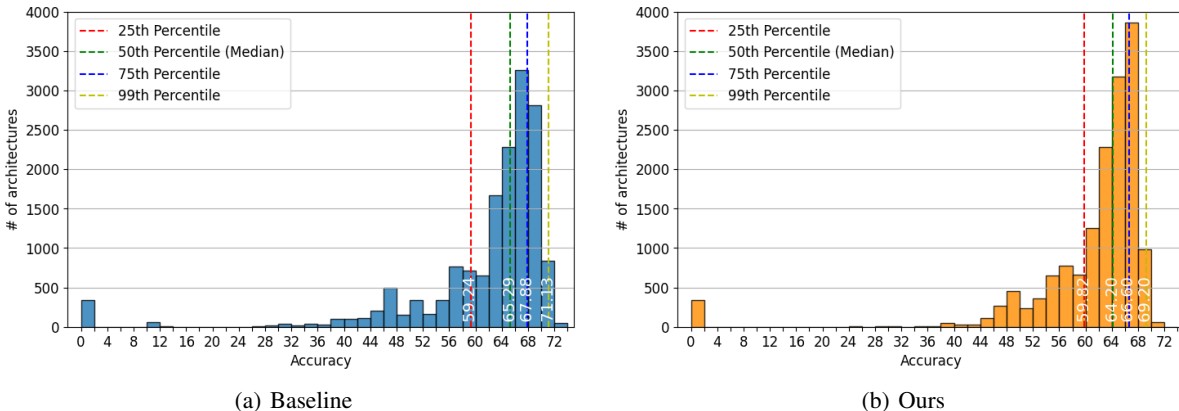

(a) Baseline           (b) Ours

*Figure 7.* Comparison of architecture distribution within the search space. 7(a) is the baseline, which is the result of training from scratch, provided by NAS-Bench-201. 7(b) is our result, where we directly evaluate the inherited weights from the supernetwork without any retraining.

We analyzed the architecture distribution within the search space of NAS-Bench-201, as shown in Figure 7. We can see that, compared to the baseline, although slightly less in the highest performance, the architecture distribution of our method is more concentrated at relatively high performance.

