# OpenReview forum: "Single Train Multi Deploy on Topology Search Spaces using Kshot-Hypernet"
_ICML.cc/2024/Workshop/WANT — WANT@ICML 2024 Poster_

### Official Review · Reviewer_b9w2 · 2024-06-12
**The paper proposes a NAS method that does not require retraining based on the topology search space**

**Confidence:** 5

**Summary:**

The paper proposes  a NAS method that does not require retraining based on the topology search space, instead of size space. The method is able to train overlapping (but not totally subordinate) sub-networks as a part of a super-network. The  method combines the advantages of previously proposed Hypernetwork and Kshot-NAS, as well as a distillation and sampling methods. Performances are shown on NAS-Bench201 and show that the method matches and possibly exceeds the baseline performance of post-search retraining.

**Strengths:**

- The paper is tackling the topic of sub-network search that may achieve efficient deployment without retraining.
- The paper proposes a sampling method with a high probability of sampling performant sub-networks while minimizing the impact on the remaining sub-networks.
- Some experiments show the impact of the proposed method against retraining.

**Weaknesses:**

# Contribution
- The proposed method is a combination of widely known/existing concepts in simultaneous super and sub-networks training: one may expect a clearer and more precise explanation of what is really novel at least in the bullet points of page 3.

# Motivation of the work
- A very important discussion of the relationships between "size search" and "topology search" is insufficient in the paper: when the former is useful instead of the latter? and vice versa?  In the sentence "Although the size search space is still a very large search space that can be effectively searched for different devices, this search space also has significant drawbacks.": the author may elaborate further about what are these drawbacks, and when one may use topology search instead (early enough in the paper).

# Reference/discussion of prior work
- Some references on training multiple networks are missing: searching sub-networks based on their topology (without retraining) is not a new concept, and related work exists and some of the most recent work is not cited in this paper. Some of the existing methods are able to define topologies that could be trained simultaneously as a part of super-network training.

# Experiments
- Experiments involve search spaces with 15625 architectures, so one may question the extension of this method to larger architectures and larger search spaces.
- Comparisons involving more datasets (and possibly other architectures) are missing which makes it difficult to judge the generalization of the proposed method to more challenging settings.

# Presentation, clarity and writing
- The presentation of the paper as well as its writing need to be substantially improved (examples below).
- In the sentence "However, as mentioned earlier, these are all based on the small search space and will not search for new
architectures, all of which are variants of MobileNet, but the methods proposed are very meaningful.":  what methods are meaningful ? in what sense? In this part of the paper, one may expect more details about the technical differences of these related methods against the author's claimed contribution.
- The sentence "Weight sharing has not been theoretically verified." is not clear ... same remark for "the ranking correlation of super-networks based on weight sharing": what is the ranking correlation? same remark for "the mutual interference" ... all these concepts (even introduced in the introduction and related work) need to be clarified and reminded ... in order to make the paper clear and easier to follow.
- Section about FocusFair sampling needs a better clarification of different steps; many statements in this subsection are not clear. For instance "higher architecture parameters β between each pair of nodes"  is not clear:  what are these pairs of nodes?
- Some figure captions are very brief, and need to be further expanded.
- Important to define acronyms at their first use: OFA, GHN, etc. Same remark for the used variables in the math.
- English usage needs to be improved:  some parts of the paper are difficult to follow and contain multiple repetitions of words and expressions (ex. lines 55 to 91; where the expression "search space" is repeated 14 times in the same paragraph). Another example in the sentence "The network is stacked from search blocks, and we can search for one architecture for each block": there is a mix of active and passive forms within a short distance in the text, and also in the sentence "so that its sub-networks can also work.": what "work"? another example "weights the K weights", etc.
- Typos and suggested updates:  "for each platforms" -> "for each platform",  "like us" -> "similarly to our method",...

---

### Official Review · Reviewer_hpDS · 2024-06-13
**Novel training paradigm**

**Confidence:** 3

**Summary:**

This paper introduces an algorithm combining Hypernetwork and KshotNAS. English writing should be improved.

**Strengths:**

Interesting combination of two existing works. Applying an adapted knowledge distillation method in sampling.

**Weaknesses:**

Writing needs to be improved. Words are often missed or repeated in a sentence.

In the conclusion the authors claim "... (our method) significantly enhancing the performance of all architectures within this search space." There is no supporting evidence in the paper since only the average performance is reported (unless the authors missed the word "average" in the sentence).

Also, I am not fully convinced about choosing the average accuracy as the performance metrics which is not a popular option among the other approaches. Although different subnets are needed for different platforms, is it really necessary to have *all* the sampled subnets achieve good performance?

Overall, the experiments lack detailed benchmarking which weakens the conclusion about the performance.

---

### Official Review · Reviewer_93RX · 2024-06-14
**Weak Accept: The paper proposes a novel super-network training method for the topological search space and shows promising results on NAS-Bench-201.**

**Confidence:** 4

**Summary:**

The paper proposes a novel super-network training technique based on the topological search space in NAS-Bench-201. The proposed technique combines ideas from the Hypernetwork and the K-Shot NAS approaches to improve the expression ability and performance. Additionally, the paper also introduces a new distillation and sampling strategy for training the super-network and avoid any additional re-training.

**Strengths:**

- The paper is very well written and easy to follow.
- The proposed approach combines ideas from Hypernetwork and K-Shot NAS in an interesting way to solve the super-network training problem for topological search spaces and avoid having to re-train sub-networks from scratch for good performance.
- The results presented in the paper for average accuracy on CIFAR10 and CIFAR100 in NAS-Bench-201 seems promising for further exploration.

**Weaknesses:**

- The paper lacks comprehensive ablation studies on compute time, memory and accuracy tradeoffs using the proposed approach. For example, there is a 1-2% gap in the max accuracy compared to the baseline. Can this gap be reduced with further training?
- In addition to reporting average and max accuracy, it might be better to also show the accuracy differences for certain selected sub-networks compared to the baseline.
- Missing results on other larger datasets and benchmarks.  The results on ImageNet-16-120 which is also part of NAS-Bench-201 is not shown in the paper.
- It is not clear if the proposed approach can be extended beyond CNNs to other architectures like transformers.

**Suggestions:**

- The authors can consider including the accuracy differences for certain selected sub-networks of different sizes compared to the baseline, instead of just reporting the average and max accuracy.
- Authors can also consider including results on ImageNet-16-120 for completeness on NAS-Bench-201

---

### Meta-Review · Area_Chair_WpoZ · 2024-06-16

**Recommendation:** Accept (Poster)
**Confidence:** 5

**Metareview:**

The paper introduces a retraining-free NAS method with respect to the topology search space. The approach aims to train an overlapping set of subnet as part of a larger super-network, combining Hypernetwork with Kshot-NAS and distillation. The AC agrees with the reviewers regarding the method's novelty. For the final version, the authors should focus on improving the writing  and incorporate the suggestions received from the reviewers.

---

### Decision · Program_Chairs · 2024-06-17

**Decision:**

Accept (Poster)

**Comment:**

We thank the authors for their time and contribution to WANT and we are pleased to share that after the reviewing process the paper has been accepted. Congratulations! We encourage the authors to consider reviewers' feedback for the improvement of the camera-ready version. We hope to see you in person at the workshop and brainstorm on efficient training research together!